Subject Area:
biochemistry/microbiology/bioinformatics/
structural biology

Keywords:
photosynthesis, photosystem, water oxidation, oxygenic, anoxygenic, reaction centre

Author for correspondence:
Tanai Cardona
e-mail: t.cardona@imperial.ac.uk

# Thinking twice about the evolution of photosynthesis

Tanai Cardona

Department of Life Sciences, Imperial College London, London, UK

(iD) TC, 0000-0003-2076-4115

Sam Granick opened his seminal 1957 paper titled 'Speculations on the origins and evolution of photosynthesis' with the assertion that there is a constant urge in human beings to seek beginnings (I concur). This urge has led to an incessant stream of speculative ideas and debates on the evolution of photosynthesis that started in the first half of the twentieth century and shows no signs of abating. Some of these speculative ideas have become commonplace, are taken as fact, but find little support. Here, I review and scrutinize three widely accepted ideas that underpin the current study of the evolution of photosynthesis: first, that the photochemical reaction centres used in anoxygenic photosynthesis are more primitive than those in oxygenic photosynthesis; second, that the probability of acquiring photosynthesis via horizontal gene transfer is greater than the probability of losing photosynthesis; and third, and most important, that the origin of anoxygenic photosynthesis pre-dates the origin of oxygenic photosynthesis. I shall attempt to demonstrate that these three ideas are often grounded in incorrect assumptions built on more assumptions with no experimental or observational support. I hope that this brief review will not only serve as a cautionary tale but also that it will open new avenues of research aimed at disentangling the complex evolution of photosynthesis and its impact on the early history of life and the planet.

## 1. The study of the evolution of photosynthesis

Anoxygenic photosynthesis pre-dates the origin of oxygenic photosynthesis [1]. After the emergence of the earliest forms of anoxygenic phototrophic bacteria, the capacity scattered across a few groups of bacteria via horizontal gene transfer [2,3]. Oxygenic photosynthesis originated in an ancestor of Cyanobacteria when an anoxygenic photosystem gave rise to a water-splitting photosystem [4]. These three basic premises currently underlie the study of the evolution of photosynthesis and would hardly make anyone raise a sceptical eyebrow.

Up for debate are the mechanisms by which Cyanobacteria obtained two distinct photochemical reaction centres linked in series, Photosystem I and Photosystem II, the hallmark of oxygenic photosynthesis. Was the origin of Photosystem I and Photosystem II triggered by a gene duplication event occurring before phototrophy scattered across the tree of life [5–7]? Or were these distinct photosystems acquired via horizontal gene transfer from lineages of anoxygenic phototrophs into a non-photosynthetic ancestor of Cyanobacteria right before the great oxidation event (GOE) [8,9]? We can therefore debate how long it took for anoxygenic photosynthesis to emerge after the origin of life, and how long it took for oxygenic photosynthesis to emerge after the origin of anoxygenic photosynthesis. We could also debate the identity of the earliest phototrophs, the oldest type of photosystem, and whether this ancestral photosystem used chlorophyll, bacteriochlorophyll or a mixture of both.

By 2007, when the discovery of phototrophic Acidobacteria was first published [10], all groups of phototrophs then known had been proposed as the innovators of phototrophy, a wide-range of bacterial cell fusion and horizontal

gene transfer combinations had been suggested for the birth of Cyanobacteria, and almost every possible scenario for the nature of the earliest photochemical reaction centre had already been put forward. I reviewed this briefly before [11] (but see also [12]). It is only proof that the study of the evolution of photosynthesis is fascinating; it exerts an almost irresistible force on the curious mind that marvels at the origin of things and drives us to speculate. I have not been able to resist this force despite my best efforts [13]; I have certainly not been the first [14] and will undoubtedly not be the last [15].

This long history of speculation on the evolution of photosynthesis has blurred the line between what is assumed to be true and the facts as supported by rigorous evidence. In this essay, I shall demonstrate that several core ideas in the study of the evolution of photosynthesis are based on unsupported—sometimes incorrect—assumptions. A comprehensive critical assessment of the molecular evolution of photosynthesis would require more space than I have here, for that reason I will only focus on these three basic premises: (i) that anoxygenic reaction centres are more 'primitive' than those in oxygenic photosynthesis, (ii) that the horizontal transfer of photosynthesis is more likely than the loss of photosynthesis, but above all (iii) that anoxygenic photosynthesis pre-dates oxygenic photosynthesis.

## 2. Primitive photosynthesis

It is widely believed that the photochemical reaction centres used in anoxygenic photosynthesis are more primitive than those used in oxygenic photosynthesis. Ten years ago, when I was still a PhD student and a few years before the study of the evolution of photosynthesis became my full-time job, I came across an interesting and very memorable analogy: that the anoxygenic Type II reaction centre of phototrophic Proteobacteria and Chloroflexi is like a Ford Model T car, and in comparison Photosystem II, the water-oxidizing enzyme, is like a Formula 1 racing car, implying that the former gave rise to the latter [16]. The logic behind the premise that anoxygenic photosynthesis is primitive goes like this: *Water oxidation to oxygen is a difficult chemical reaction that requires great complexity. Photosystem II is more complex than anoxygenic Type II reaction centres; so surely, the anoxygenic reaction centre is more primitive and must have given rise to the oxygenic one.*

This assumption is not new and its roots can be traced back to speculative commentary emerging from the early comparative biochemistry of anoxygenic and oxygenic photosynthesis, starting more than 80 years ago [17–19]. For example, H. F. Blum in 1937 reasoned that because oxygenic photosynthesis used four quanta of light to complete a catalytic reaction, unlike anoxygenic photosynthesis, the latter must have been more primitive [19]. Then, this idea was reworked, cemented and popularized by Olson [20], whose insight into the evolution of photosynthesis is as influential today as it was back then [21]. It should be noted that this idea became popular well before we had a comprehensive understanding of the photosynthetic processes and long before we had access to sequences or structures of the reaction centres to put it to the test. Now this assumption has taken the appearance of almost undisputable fact.

While at first glance it seems quite reasonable, the flaw is found in the incorrect presupposition that the complexity of oxygenic photosynthesis, and by extension of Photosystem II, evolved *before* the origin of water oxidation photochemistry. Perhaps it is not too counterintuitive to think that the origin of water oxidation should be considered the trigger that led to an increase in complexity, because the increased complexity exists for the sole purpose of supporting water oxidation. *After* the origin of water oxidation greater complexity was evolved to make catalysis more robust and efficient [22,23], to incorporate protection mechanisms against the formation of reactive oxygen species and to diminish the risk of damage [24,25]. Increased damage caused by reactive oxygen species led to the evolution of a more complex repair and assembly process [26,27]. An increase in structural and functional complexity also led to the evolution of more sophisticated regulatory processes acting from picoseconds to weeks and ranging from fine-tuning of electron transfer [28–30] to long-term chromatic adaptation [31]. The bottom line is that an apparent lack of complexity is not a definitive measure of primitiveness, yet that apparent complexity of Photosystem II disappears when the core reaction centre proteins are compared with each other (figures 1 and 2).

The assumption that anoxygenic reaction centres are more primitive than those used in oxygenic photosystems is not supported by phylogenetic and structural evidence [11,32]. Of note is the fact that the heterodimeric core of Photosystem II, made up of D1 and D2, originated from an unambiguous gene duplication event distinct to that which led to the heterodimeric core of the anoxygenic Type II reaction centres, made of L and M [11]. The reason why this is unambiguous lies in the fact that D1 and D2 share much greater sequence and structural identity with each other than with L and M; and vice versa. While all Type II reaction centre proteins share common ancestry, the known anoxygenic Type II reaction centres are not direct ancestors of Photosystem II. They cannot be described as more primitive and they do not make any better models for what ancestral photosystems looked like than Photosystem II. The assumption that the ancestral Type II reaction centre before L, M, D1 and D2, was more like those found in Proteobacteria and Chloroflexi than those in Cyanobacteria, carries a wealth of unproven assumptions: for example, that L and M retained more ancestral traits than D1 and D2, or that L and M are evolving at a significantly slower rate than D1 and D2.

The case for Type I reaction centres is similar. Anoxygenic Type I reaction centres are homodimeric, while Photosystem I in oxygenic photosynthesis is heterodimeric. Undoubtedly, the homodimeric state is the ancestral state, but that does not necessarily imply that Photosystem I in oxygenic photosynthesis directly originated from the reaction centre of any of the known groups of anoxygenic phototrophs. The phylogeny of Type I reaction centres indicates that all anoxygenic Type I homodimers share a more recent common ancestor to the exclusion of Photosystem I core subunits [11], which is also a reflection of the greater sequence and structural identity among homodimeric Type I reaction centres [33]. It is indeed correct to say that a heterodimeric core is a novel trait relative to the ancestral state, but it would be incorrect to say that PshA or PscA, the reaction centre core subunits of phototrophic Firmicutes, Chlorobi and Acidobacteria, gave rise to the ancestral core subunit of cyanobacterial Photosystem I.

Cardona *et al.* recently showed that the core subunits of the anoxygenic Type II reaction centres are evolving on

royalsocietypublishing.org/journal/rsob Open Biol. 9: 180246

royalsocietypublishing.org/journal/rsob   Open Biol. 9: 180246

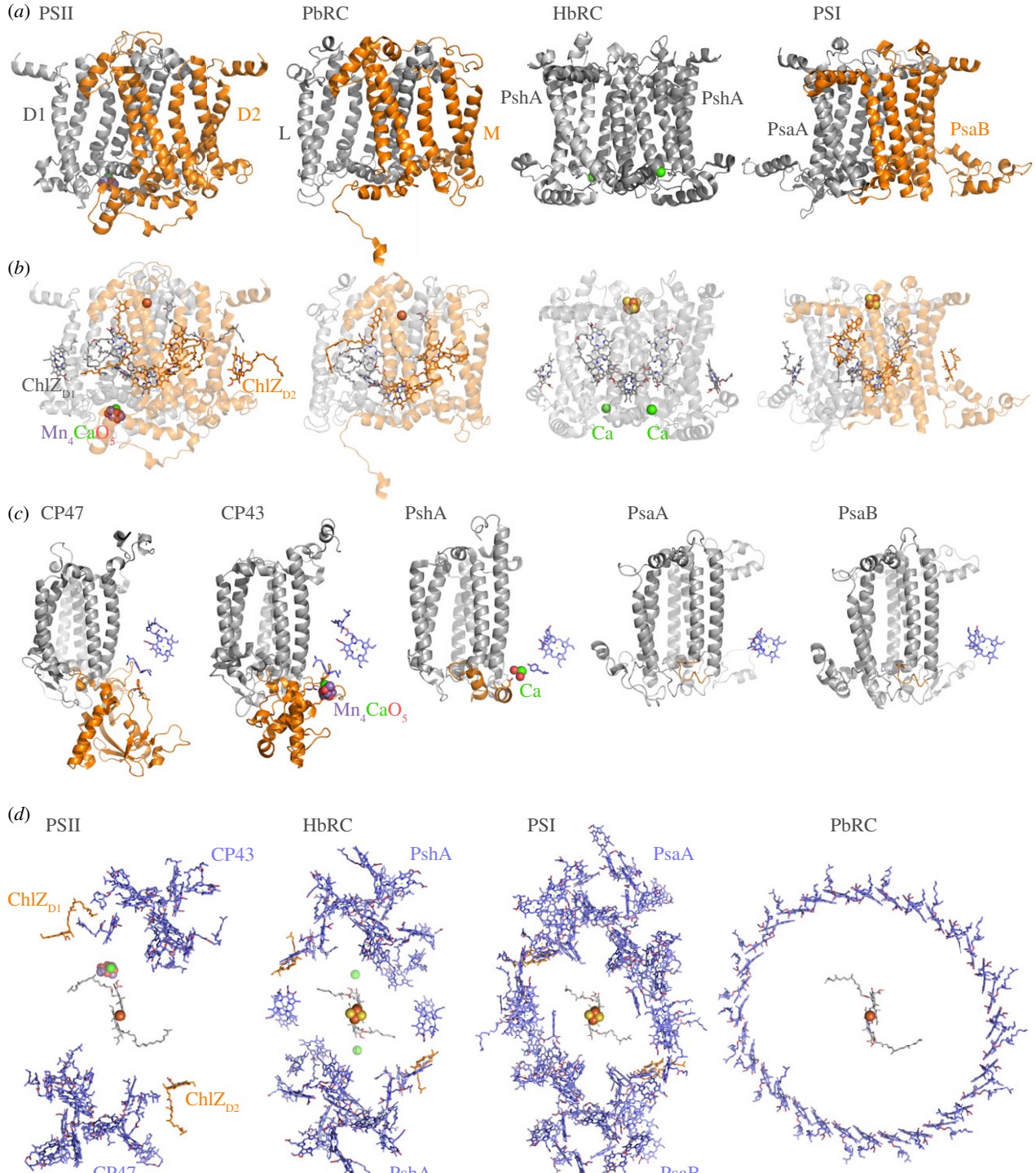

**Figure 1.** Structural comparisons. (*a*) The reaction centre core subunits. PSII stands for cyanobacterial Photosystem II, PbRC for proteobacterial reaction centre, HbRC for heliobacterial reaction centre and PSI for cyanobacterial Photosystem I. (*b*) The core subunits are shown transparently to highlight the photochemical pigments. Type II reaction centres are characterized by a quinone/non-heme $Fe^{2+}$/quinone electron acceptor system. Type I reaction centres are characterized by a $Fe_4S_4$ cluster electron acceptor system. $ChlZ_{D1}$ and $ChlZ_{D2}$ are a pair of peripheral chlorophylls bound to the core, but functionally associated with the antenna domain. These are conserved in PSII and other Type I reaction centres and are absent in anoxygenic Type II. (*c*) The antenna domains: CP43 and CP47 are the antenna of PSII, PshA of the HbRC and PsaA/PsaB of PSI. In Type I reaction centres, the core and the antenna make a single protein. In PSII, the core and the antenna are separate proteins. Anoxygenic Type II reaction centres (PbRC) lack antenna domains, but have independently evolved a new light-harvesting complex. The $Mn_4CaO_5$ cluster is bound by D1 and CP43. In the HbRC, a Ca is found at a similar position and it is bound by the core and the antenna as in PSII. (*d*) A global top view of all the (bacterio)-chlorophyll light-harvesting pigments. $ChlZ_{D1}$ and $ChlZ_{D2}$, and their equivalent in Type I reaction centres, are shown in orange. The other antenna pigments are shown in blue. Which of these structures is the most primitive? PDB ID: PSII, 3wu2; PbRC, 5y5 s; HbRC, 5v8 k; PSI, 1jb0.

average about five times faster than the core subunits of the water-oxidizing enzyme [34]. They have probably done so for most of their evolutionary history. That D1 and D2 are evolving significantly slower has two major implications: (i)

that the duplication of the oxygenic core that led to D1 and D2 is older than the duplication that led to L and M; and (ii) that Photosystem II has retained more ancestral traits than its anoxygenic cousin. This is clearly seen in the

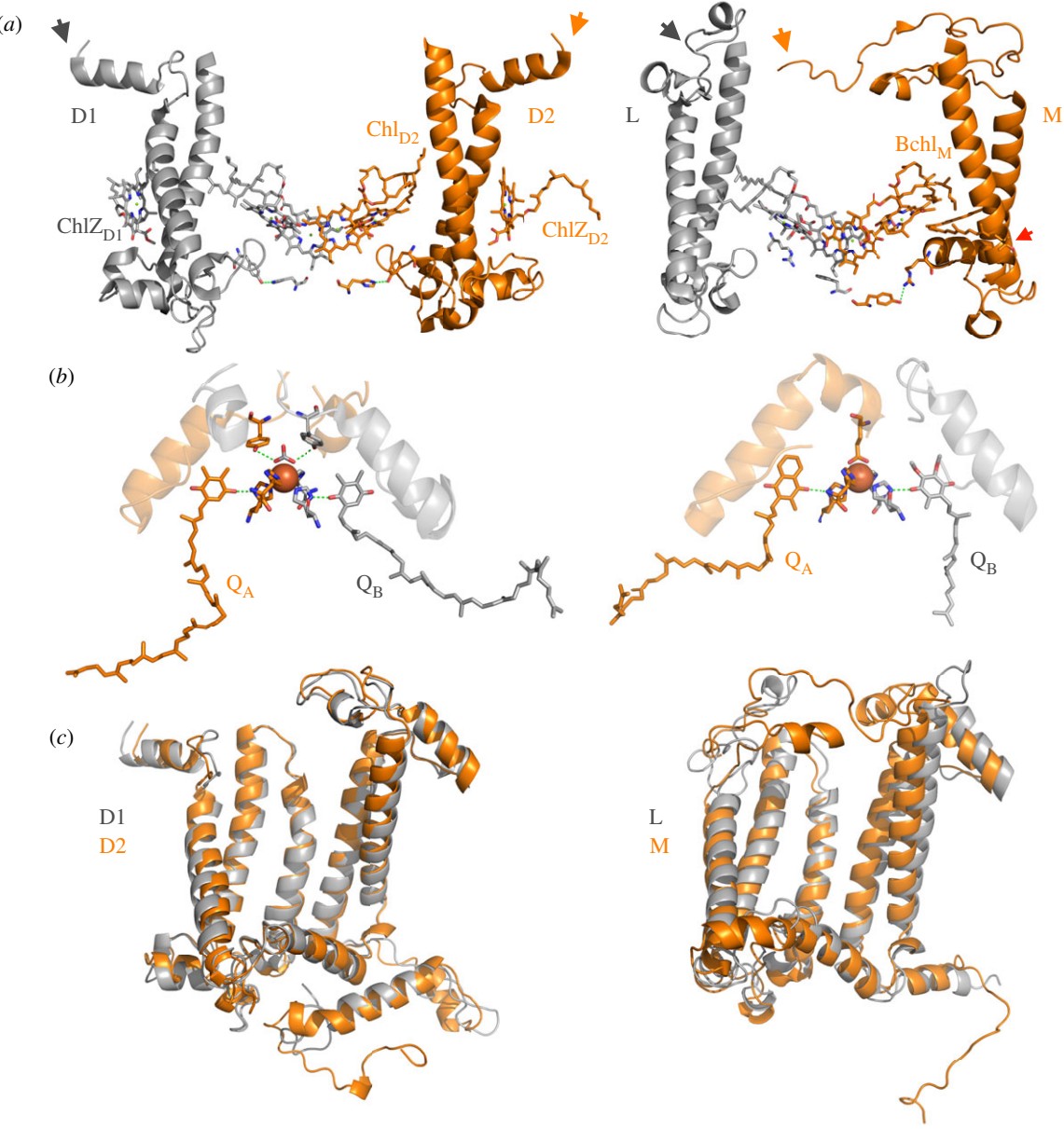

**Figure 2.** Heterodimerization of Type II reaction centres. The core of Photosystem II has retained more symmetry than anoxygenic Type II reaction centres. (*a*) A comparison of some structural elements between Photosystem II and the proteobacterial anoxygenic Type II reaction centre (PbRC). Only the first two transmembrane helices of the core subunits are shown. The N-terminus is marked with a grey and an orange arrow: there is noticeably more symmetry at the N-terminus between D1 and D2 than between L and M. More symmetry in Photosystem II is also observed around the overlap of the 4th transmembrane helix, and the C-terminus. In the PbRC, the bacteriochlorophyll peripheral pigment Bchl$_M$ interacts with an asymmetrically located carotenoid. This is not found in the reaction centre of the Chloroflexi, instead a 'third' pheophytin takes the position of Bchl$_M$ conserving the pigment asymmetry. In Photosystem II, strictly conserved redox-active Tyr-His pairs are found on D1 and D2 at the donor side. In the PbRC, an Arg takes the position of Tyr. In M, R164 provides a hydrogen-bond to Y193, which is substituted by Phe in the L subunit. (*b*) A comparison of symmetry at the quinone/non-heme Fe$^{2+}$/quinone electron acceptor system. In Photosystem II, the non-heme Fe$^{2+}$ is coordinated by bicarbonate, which is symmetrically bound by strictly conserved Tyr residues. In the PbRC, the non-heme Fe$^{2+}$ is coordinated by a Glu residue in the M subunit showing that the PbRC electron acceptor side cannot have retained the ancestral state. (*c*) Overlap of D1 and D2 (left) and L and M (right). The root-mean-square deviation of atomic positions (RMSD) is a measurement of average distance between overlapped atoms of the backbone's alpha carbons. Between D1 and D2 the RMSD is 2.32 Å over 320 residues, while between L and M is 4.37 Å over 232 residues. The greater the symmetry, the better the overlap and the smaller the RMSD over a greater number of residues.

structures of the photosystems, as Photosystem II has retained not only greater sequence and structural symmetry at the core but also greater structural identity with Type I reaction centres (figures 1 and 2). In a manner similar to Type II reaction centres, by studying the rates of evolution I have also found that the gene duplication leading to the heterodimeric core of cyanobacterial Photosystem I has a pretty good chance of having occurred before the diversification event leading to the different groups of phototrophs with homodimeric Type I reaction centres known today [35].

In conclusion, the assumption that anoxygenic reaction centres are more primitive than those used in oxygenic photosynthesis is, at best, unsupported by their molecular evolution; at worst, it is incorrect.

## 3. Easy transfer

Arguably there is no topic more controversial in the study of the evolution of photosynthesis than whether the scattered

royalsocietypublishing.org/journal/rsob   Open Biol. 9: 180246

distribution of phototrophy is largely due to widespread horizontal gene transfer or multiple losses across the tree of life. But why is this important? If horizontal transfer of photosynthesis is a relatively easy process, then one could argue that oxygenic photosynthesis could have emerged at a late stage in the evolutionary history of life from the transfer of anoxygenic photosynthesis into a non-photosynthetic ancestor of Cyanobacteria, for example. If, on the other hand, the transfer of photosynthesis is a more difficult process then it may be more likely that the emergence of two distinct photosystems was the result of a gene duplication event, which occurred in a deep but direct ancestor of Cyanobacteria. Certainly, gains and losses of photosynthesis across geological time are not mutually exclusive, and there is evidence that both have occurred.

This leads to the next popular assumption in the study of the evolution of photosynthesis: the idea that the probability of a non-photosynthetic organism gaining photosynthesis via horizontal gene transfer is greater than that of a photosynthetic organism losing photosynthesis. Given this assumption, if a clade of photosynthetic bacteria shares a more recent common ancestor with a non-photosynthetic clade, it is *assumed* that the ancestral state is more likely to be non-photosynthetic [2,3]. The recent discovery of early-branching non-photosynthetic Cyanobacteria, the Melainabacteria [36] and the Sericytochromatia [9], has lent credence to the hypothesis that oxygenic photosynthesis was invented after horizontal transfer of anoxygenic photosynthesis. This hypothesis relies on the *assumption* that the ancestral state was non-photosynthetic.

To the best of my knowledge, there are no published studies that have attempted to determine the likelihood of gain versus loss of photosynthesis across the tree of life.

There is only one case of a group of non-photosynthetic bacteria obtaining photosynthesis via horizontal gene transfer. That is the phototrophic Gemmatimonadetes. Zeng *et al.* demonstrated that this peculiar group of bacteria obtained a photosynthesis gene cluster from a gammaproteobacterium [37]. Any other case of potential gains of photosynthesis in bacteria is ambiguous. The other (more spectacular) case of gain of photosynthesis are the photosynthetic eukaryotes. Horizontal gene transfer occurred from a cyanobacterium endosymbiont into the host nuclear genome of a non-photosynthetic unicellular eukaryote [38,39]. Nevertheless, after more than a billion years of evolution and the transfer of hundreds, if not thousands [40], of cyanobacterial genes into the eukaryotic nuclear genome, the core subunits of the photosystems have remained encoded in the plastid genome.

There are many clear cases of horizontal gene transfer of photosynthesis components *between* phototorphs. Several studies have shown, for example, that the marine *Synechococcus* and *Prochlorococcus* strains obtained protochlorophyllide reductase from Gammaproteobacteria [5,41]. Their genes are distinctly and undoubtedly proteobacterial as they cluster specifically within Proteobacteria. Another peculiar case is the transfer of protochlorophyllide and chlorophyllide reductases between a stem-group phototrophic Chlorobi and a stem-group phototrophic Chloroflexi [42], but the direction of transfer is ambiguous.

Recently, it was suggested that it was not merely Cyanobacteria which obtained photosynthesis via horizontal gene transfer: a compelling case has been made for phototrophic Chloroflexi evolving via the transfer of photosynthesis as early as approximately 900 Ma [43,44]. Gisriel *et al.* also

proposed that Heliobacteria, the only known phototrophic Firmicutes, also obtained phototrophy via horizontal transfer justified by the presence of a photosynthesis gene cluster [45]. It is noteworthy that Cyanobacteria and Chloroflexi show phylogenetic affinity in many phylogenomic analyses appearing in many instances as each other's closest relatives [46–50]. One has reasons to argue that the most recent common ancestor (*mrca*) of these two phyla had Type II reaction centres. Indeed, the large phylogenetic distance between the core subunits of the anoxygenic Type II reaction centre and Photosystem II would be consistent with such a scenario. Similar kind of reasoning can be presented against the hypothesis that Heliobacteria obtained phototrophy via horizontal gene transfer. For example, Mix *et al.* noted that the evolution of Type I reaction centres is consistent with vertical descent [51], and if recent phylogenomic trees of prokaryotes have any resemblance to reality only a single gain of horizontal gene transfer of photosynthesis between phyla of bacteria can be identified [52]: phototrophic Gemmatimonadetes, the exception that proves the rule (figure 3).

An often-cited piece of evidence in favour of horizontal gene transfer scenarios is the fact that in some anoxygenic phototrophs most of the genes required to support phototrophy are encoded in a single gene cluster. Recently, Brinkmann *et al.* showed that within the family Rhodospirilaceae of Alphaproteobacteria the transfer of complete photosynthesis gene clusters has occurred [53]. The authors calculated that even in the most stringent scenarios at least seven transfer events and eight loss events were needed to reconcile the species tree with the tree of the gene cluster. If these numbers are accurate that would make losses slightly more probable than gains, at least within the Rhodospirilaceae. However, from these seven transfers at least five can be better described as replacements of a native gene cluster with that from a very closely related strain rather than true *de novo* gains of phototrophy, making the probability of losses substantially greater than gains. It should be noted however, than in none of these transfer cases the photosynthesis gene cluster originated from outside the Rhodospirilaceae. Thus, it could be argued that the greater the phylogenetic distance between the donor and the recipient strain, the less likely it will be that a non-phototroph will successfully integrate and express an entire photosynthesis gene cluster.

We can now wonder if the reason why phototrophic Gemmatimonadetes managed to successfully integrate a photosynthesis gene cluster from Proteobacteria is because the phylum may have been ancestrally phototrophic to begin with, given that Gemmatimonadetes and Chlorobi show strong phylogenetic affinity [50,52]. This seems unlikely if one is biased towards thinking that lack of photosynthesis is a more plausible ancestral state. A relevant example of these exchange processes was recently reported in the dinoflagellate *Lepidodinium*. It was shown that its ancestor lost photosynthesis, discarded the entire pathway for the synthesis of chlorophyll *a*, then regained photosynthesis by acquiring a new algal endosymbiont, and finally rebuilt the chlorophyll synthesis pathway with genes transferred from multiple sources, rather than from the new endosymbiont itself [54].

There are many examples of losses of oxygenic photosynthesis in Cyanobacteria and eukaryotes. One fascinating case is the loss of all Photosystem II-encoding genes in *Atelocyanobacterium thalassa* (UCYN-A), along with the loss of roughly 75% of the original genome content [55] in only

royalsocietypublishing.org/journal/rsob  Open Biol. 9: 180246

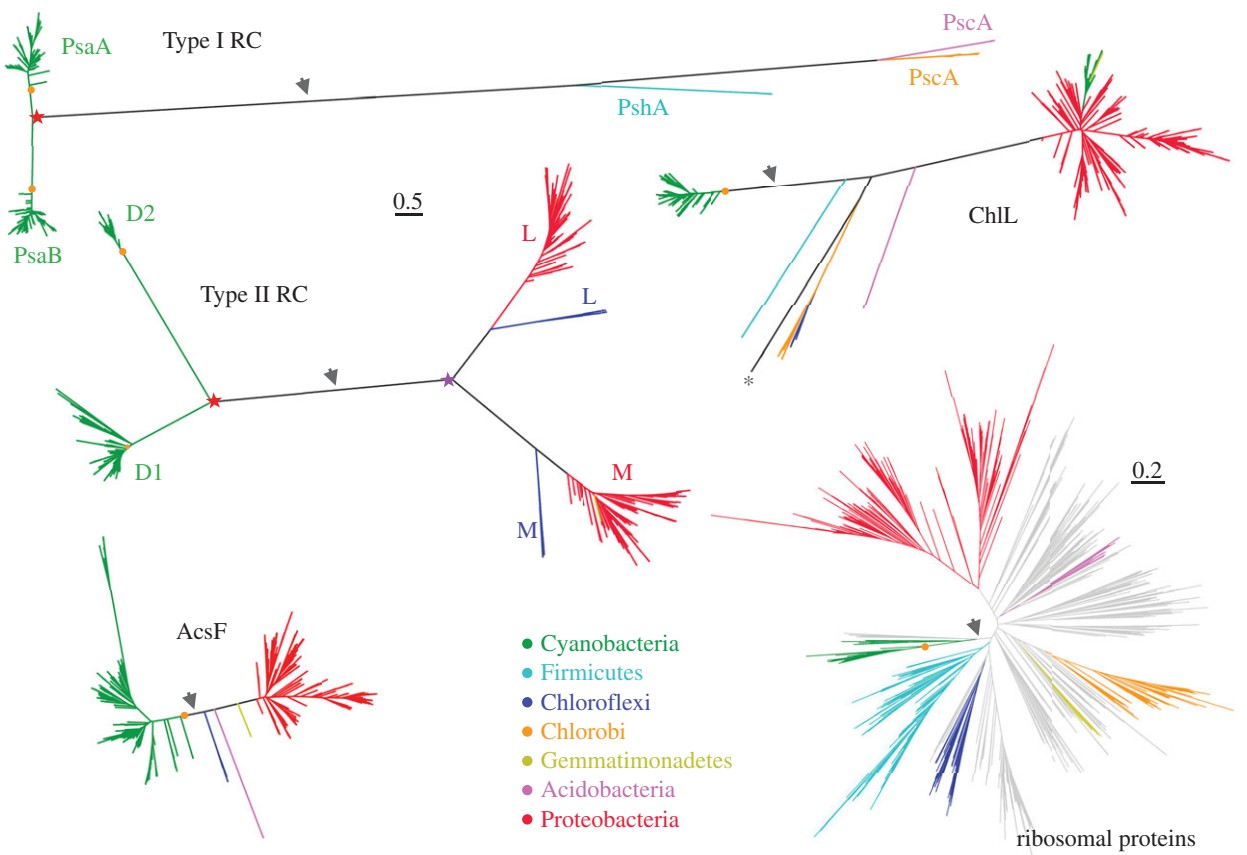

**Figure 3.** Loss of photosynthesis or horizontal gene transfer? Maximum-likelihood trees of Type I reaction centre proteins, Type II reaction centre proteins, protochlorophyllide reductase subunit L (ChlL), oxygen-dependent Mg-protoporphyrin IX monomethyl ester cyclase (AcsF) and a tree of concatenated ribosomal proteins reported before [52]. The different phototrophic clades are coloured at phylum level. The topology of the trees of photosynthetic components matches closely the tree of ribosomal proteins subtracting non-phototorphic clades and accounting for duplication events (stars). The grey arrow represents the position of the root of the tree as calculated in [52]. The orange circle marks the most recent common ancestor of Cyanobacteria capable of oxygenic photosynthesis (Oxyphotobacteria). The duplications leading to PsaA and PsaB, and to D1 and D2 are highlighted with a red star, and these duplications must pre-date the diversification of described Cyanobacteria. Similarly, the duplication leading to L and M (purple star) must pre-date the origin of phototrophic Chloroflexi and Proteobacteria. The asterisk marks a ChlL subunit from an uncharacterized group of phototrophs. Scale bar represents substitutions per site. All photosynthesis proteins are plotted at the same scale.

100 Myr [56]. Another interesting case of loss of photosystem genes is that of the cyanobacterium endosymbiont of rhopalodiacean diatoms [57]. In photosynthetic eukaryotes we find the well-known case of parasitic apicomplexa [58], multiple losses in dinoflagellates [59], chrysophytes [60], free-living green algae [61,62] and holoparasitic angiosperms [63], among probably many others. Within flowering plants, about 10 independent losses of photosynthetic capacity have already been documented [64]. One could argue that it is much more likely to lose a single anoxygenic photosynthesis gene cluster than to lose the entire complexity of oxygenic photosynthesis.

Not a single case of horizontal gene transfer of oxygenic photosynthesis between bacteria has been documented. Unlike anoxygenic photosynthesis, the numerous genes now required to support oxygenic photosynthesis are scattered across the cyanobacterial genomes [6] making the probability of a non-photosynthetic bacterium acquiring oxygenic photosynthesis via horizontal gene transfer nil. In contrast, and as we saw in the previous paragraph, the probability of loss of oxygenic photosynthesis is certainly above zero and can occur within less than 100 Ma. Therefore, if the probability of loss of photosynthesis is just slightly greater than gain, it is expected that over billions of years the number of lineages of anoxygenic and, in particular, of oxygenic phototrophs, has

decreased. This not only explains the scattered distribution of photosynthesis in bacteria but also the large phylogenetic distance between phototrophic lineages, a distance that is matched by that of their photosynthetic machinery (figure 3): an aspect that is always overlooked and unaccounted for in horizontal gene transfer scenarios.

This line of thought reveals another assumption on the study of the evolution of photosynthesis not based on any piece of scientific evidence, but nonetheless taken for granted: that Cyanobacteria are the only group of bacteria alive today to have descended from oxygenic phototrophs. In fact, it takes a very simple mental exercise to demonstrate that most of the diversity of oxygenic phototrophs that have existed in the history of the planet probably pre-dates the *mrca* of Cyanobacteria.

All Cyanobacteria share an *mrca* that was capable of oxygenic photosynthesis [4] in the same way that birds originated from an *mrca* that had already evolved feathers [65,66]. All Cyanobacteria share an *mrca* that had already evolved Photosystem I with a heterodimeric core. This is a trait that has been retained by *all* oxygenic phototrophs. A heterodimeric Photosystem I is a distinctive and exclusive characteristic of oxygenic photosynthesis and it is widely accepted that the heterodimerization of the core was an adaptation to oxygenic photosynthesis [29,67,68]. PsaA and PsaB,

the core subunits of Photosystem I share about 43% sequence identity: this is true across the entire diversity of oxygenic phototrophs from the earliest-branching Cyanobacteria to the most exotic variety of avocado, which implies that the *mrca* of Cyanobacteria inherited a heterodimeric Photosystem I that had PsaA and PsaB with about 43% identity. In other words, at the time of the *mrca* of Cyanobacteria, PsaA and PsaB had already changed by 57%. If we compare the change of sequence identity of PsaA across all oxygenic phototrophs, the maximum level of change is not greater than about 30%, and between all photosynthetic eukaryotes not greater than 20%. It is exactly the same for PsaB. Therefore, most of the sequence change of the core subunits of Photosystem I occurred between the time of the duplication leading to PsaA and PsaB and the *mrca* of Cyanobacteria [35]. Given that the rates of evolution of complex molecular systems is slow relative to speciation rates [69], that distance between PsaA and PsaB, that amount of change, must have been matched by a substantial biodiversity.

Such a simple exercise exposes the inherent naivety of every proposed evolutionary scenario for the evolution of photosynthesis (including my very own), since it is impossible not to severely underestimate the biodiversity of anoxygenic and oxygenic phototrophic bacteria that have existed through geological time [70]. With all of this in mind, and based on the current state of knowledge, the assumption that multiple losses of photosynthesis across the tree of life is less parsimonious than gains via horizontal transfer is far from proven, and might just as well not stand up to scrutiny.

## 4. Let there be light

The chief unproven assumption in the evolution of photosynthesis is that the origin of anoxygenic photosynthesis predates the origin of oxygenic photosynthesis. This assumption can be understood at two fundamental levels: (i) that oxygenic phototrophs evolved from anoxygenic phototrophs or (ii) that Photosystem II evolved from an anoxygenic photosystem. The first fundamental level leads to the obvious question of when Cyanobacteria originated, but the question itself is problematic. Only one point in time in the evolution of oxygenic photosynthesis can be clearly defined and accessed through phylogenies of species trees: that is, the *mrca* of Cyanobacteria (figure 4). A broad range of ages for this ancestor have been provided using molecular clocks ranging from 1.5 to 3.5 billion years [8,72–75]. Yet, in the same way that determining when the *mrca* of birds occurred cannot tell us when or how feathers originated, determining when the *mrca* of Cyanobacteria occurred cannot tell us when or how photochemical water oxidation to oxygen originated. And, in the same way that having feathers does not make *Tyrannosaurus* a modern bird, there is a very real possibility that the origin of photochemical water oxidation to oxygen does not necessarily fall in a lineage that was the immediate ancestor of the known diversity of Cyanobacteria.

A more precise line of enquiry is to determine for how long water oxidation existed before the *mrca* of Cyanobacteria. Contrary to what may appear at a first glance, the answer to this question does not depend on the exact timing of the *mrca* of Cyanobacteria or whether this ancestor existed before or after the GOE, but instead it is strongly linked to when the earliest stages in the evolution of Photosystem II occurred. For example, the duplication leading to the alpha and beta subunit

of ATP synthase is known to have occurred before the last universal common ancestor (LUCA) [76–78]. It follows then that the initial stages in the evolution of ATP synthases is in no way dependent on the time of origin of any particular group of prokaryotes (e.g. Cyanobacteria, Firmicutes, Asgardarchaeota, Euryarchaeota), but it only depends on the molecular events at play during the emergence of the protein complex itself, including the duplication leading to the alpha and beta subunits: an event that pre-dates the divergence of the domains Bacteria and Archaea, and the divergence of F-type and V-type ATP synthases.

This is also the case for oxygenic photosynthesis (figure 4). The two duplication events, which resulted in the evolution of a heterodimeric Photosystem II, one leading to D1 and D2 and the other to the core antenna subunits, CP43 and CP47, are much more likely to have occurred soon after the origin of the earliest reaction centres than right before the *mrca* of Cyanobacteria [34,35]. These two duplications together with the duplication leading to the heterodimeric core of Photosystem I are, to the best of our knowledge, exclusive to oxygenic photosynthesis [79–81]. It was suggested before based on conserved symmetrical structural and functional characteristics of Photosystem II that water oxidation started before the duplication leading to heterodimerization [34,80,81]. Cardona *et al.* recently calculated that the span of time between the duplication leading to D1 and D2 and the *mrca* of Cyanobacteria could comfortably be more than a billion years even accounting for the large uncertainties inherent to deep-time molecular clocks (figure 4) [34]. We also showed that the photosystem existing before the duplication leading to D1 and D2 had already evolved protective mechanisms to prevent the formation of singlet oxygen [34], such as bicarbonate-mediated redox tuning of electron transfer at the acceptor side (figure 2) [24]. Furthermore, our data also indicated that the span of time from the origin of photosynthesis to well past the point of duplication of D1 and D2 could be less than 200 Ma. In consequence, the only way to find out how and when oxygenic photosynthesis originated is to resolve what happened during the early evolution of photochemical reaction centres, a time in the history of life that in all likelihood considerably pre-dates the appearance of the taxonomic group that today we recognize on a 16S RNA basis as Cyanobacteria.

The second fundamental level of understanding relies on the assumption that anoxygenic reaction centres are more primitive than Photosystem II: as I discussed before, this assumption is incorrect. And all of a sudden, we find ourselves in a situation in which the fundamentals of the evolution of photosynthesis are based on unproven assumptions built on top of unproven assumptions. For instance, the idea that anoxygenic reaction centres are more primitive leads to the presupposition that primordial reaction centres did not have enough oxidizing power to split water, which then makes it easy to *assume* that water oxidation could not be an ancestral trait. An assumption that has not been, in any way, validated. Now, these ideas have such a strong grasp on our current understanding of the evolution of life that conceiving a highly oxidizing water-splitting photosystem as the primordial photochemical ancestor, and as part of the bioenergetics toolkit of early life, becomes unthinkable [82], despite the absence of unequivocal data supporting otherwise.

From a palaeobiological and geochemical perspective, the timing of the origin of oxygenic photosynthesis is far from settled. Rocks older than 2.5 billion years make less than 5%

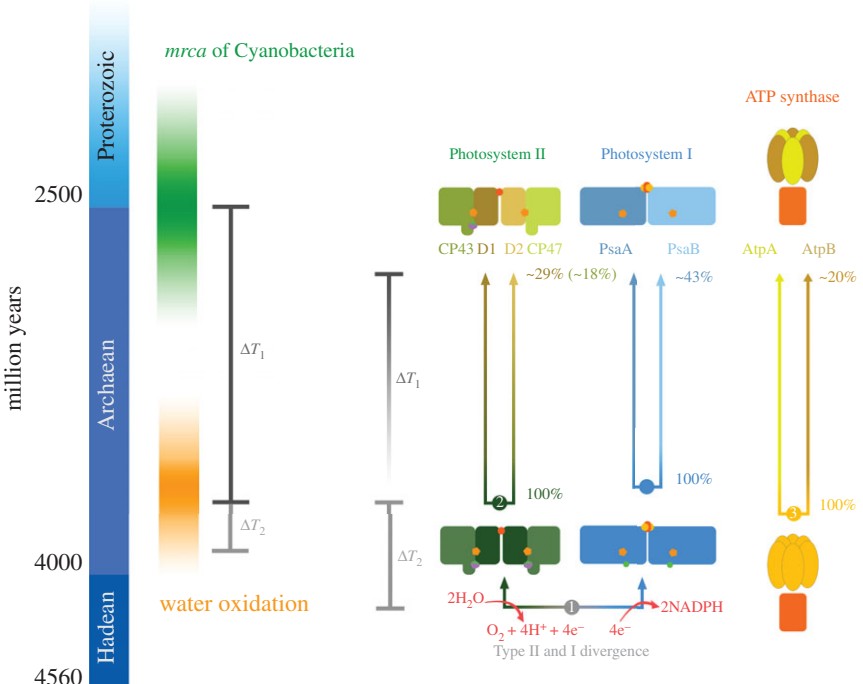

**Figure 4.** Schematic of the evolution of photosynthesis. Two key transitional points are highlighted: the divergence of Type II and Type I reaction centres, marked **1**; and the duplication leading to D1 and D2, marked **2**. We can then define two periods of time: one starts with the duplication of D1 and D2 and ends with the emergence of standard Photosystem II ($\Delta T_1$), which was inherited by the *mrca* of Cyanobacteria; and the other one starts with the Type II/Type I divergence and ends with the duplication of D1 and D2 ($\Delta T_2$). Cardona *et al.* determined that $\Delta T_1$ could comfortably be 1 billion years or more [34]. The main reason for this is the very slow rates of evolution of Photosystem II in Cyanobacteria and photosynthetic eukaryotes. Even when $\Delta T_1$ is 1 billion years the rate of evolution at the moment of duplication is calculated to be about 40 times greater than any rate observed in Cyanobacteria or photosynthetic eukaryotes. It also requires an exponential decrease in the rates approaching current rates before the Archaean/Proterozoic transition. Furthermore, due to the exponential decrease in the rates, $\Delta T_2$ can be well under 200 million years. It is worth comparing to the evolution of ATP synthases as both display indistinguishable evolutionary trends, both showing similar rates of evolution and a high degree of sequence conservation across distant taxa. The duplication leading to AtpA and AtpB, marked **3**, is known to have occurred in the LUCA. At the moment of duplication, the sequence identity between these two was 100%. Today and across all life (including Cyanobacteria), the level of sequence identity of AtpA and AtpB is about 20%. It stands to reason that the span of time between the AtpA/AtpB duplication and the F-type ATP synthase of the *mrca* of Cyanobacteria is in the order of a billion years or more. In fact, an exponential decrease in the rates of evolution of the same magnitude as in Photosystem II is required for that span of time to be as large. In Photosystem II, the level of sequence identity between D1 and D2 is about 29% and that between CP43 and CP47 is about 18%. Now, if $\Delta T_1$ is proposed to be very small, that would require faster rates at the moment of duplication than any rate reported for these types of highly conserved and very slow evolving complexes. If $\Delta T_1$ is 100 million years the rate at the duplication would need to be almost 300 times greater than any observed rate: that is twice as fast as the rate of evolution of short peptide toxins of poisonous animals [71], which are usually less than 20 residues long and are among the fastest evolving proteins in biology. In comparison, the 'simplest' reaction centre [45], that of Heliobacteria, has a core protein 608 residues long. This 'simplest' of reaction centres is made up of 4 interacting protein subunits, 54 (bacterio)chlorophyll pigments, two carotenoids, a $Fe_4S_4$ cluster, 4 calcium ions and a bunch of lipid molecules. Due to the structural complexity of the bioenergetics machinery, including the photosystems and ATP synthase, it is likely that they have maintained relatively slow rates of evolution across their entire evolutionary history.

of the surviving rock mass [83]. Stromatolites, microbial mats and bacteria-like fossils 'are present throughout virtually all of the known geological record' [84], but anything older than 2.0 billion years cannot confidently be ascribed to any particular type of phototroph, although they have traditionally been described as 'Cyanobacteria-like' [84,85]. Of note are the 3.22-billion-year-old fossilized microbial mats of the Barberton Greenstone Belt in South Africa, which in many ways resemble modern cyanobacterial mats [86]. The record of carbon isotopes fractionation extending to the oldest sedimentary rocks is somewhat inconclusive on the matter as rubisco-derived $\delta^{13}C$ signatures from anoxygenic and oxygenic phototrophs overlap. Nonetheless, and as better said by Nisbet *et al.* [87], 'In many cases Archaean carbon isotopic results have been taken to demonstrate that the material is the product of an oxygenic biosphere' (p. 313, citing [88,89]). Furthermore, the interpretation of redox proxies has provided estimates for the origin of oxygenic photosynthesis through the Archaean and down

to some of the oldest rocks more than 3.7 billion years old [90–101]. Mass-independent fractionation of sulphur isotopes in sedimentary rock has constrained the concentration of oxygen to below $10^{-5}$ of the present atmospheric level (PAL) until about 2.4 billion years ago [102–104]; yet variation in the sulphur isotope record has led researchers to suggest that concentrations of oxygen probably fluctuated greatly across time and space during the Archaean [100,105,106]. To top it all, the biogenicity, indigenousness and syngenicity of the earliest rocks and signs of life have been strongly contested [107]. A perfect and recent example of this is the 3.7-billion-year-old fossilized stromatolites reported by Nutman *et al.* [108], followed by a rebuttal [109].

Molecular evolution is equally inconclusive on the timing of the origin of biological oxygen production. For example, reconstructions of the proteome of the LUCA, often pictured as an anaerobe, have retrieved the core subunit of *bona fide* terminal oxygen reductases [77,110]; a result that seems to

royalsocietypublishing.org/journal/rsob Open Biol. 9: 180246

be supported (not without controversy [111]) by phylogenetic analysis of the same enzymes [112,113]. Reconstructions of the LUCA and phylogenetic analysis also appear to indicate an early origin of superoxide dismutase, rubrerythrin and peroxiredoxin among few other oxygen-using enzymes [77,110,114–116]. Oxygen-tolerant hydrogenases [117] and cytochrome $b_6f$ complexes incorporating protective mechanisms against the formation of reactive oxygen species have also been suggested to substantially pre-date the *mrca* of Cyanobacteria [118]. To top it all, it was recently suggested based on the physico-chemical properties of amino acids that the establishment of the universal genetic code, which should pre-date the LUCA, required 'biospheric molecular oxygen' [119]. And this is not meant to be an exhaustive list. One is then forced to either dismiss the above as artefacts of one sort or another [82,110]; to accept (rather reluctantly in my personal case) that somehow the vanishingly small traces of oxygen expected from abiotic process alone, up to eight orders of magnitude below $10^{-5}$ PAL [120], played any role *at all* in the early evolution of life [113,121]; or alternatively, one must explain away each observation with a series of rationalizations well suited to each particular case.

Efforts to time the evolution of prokaryotes or the emergence of the known groups of phototrophs have also generated puzzling results. For example, the most recent common ancestors of phototrophic Firmicutes [49,122], of phototrohic Chlorobi [49,73,123], of phototrophic Chloroflexi [43,49,73] and of phototrophic Proteobacteria [8,47,49,74] have all been timed at about the GOE, but more often than not, after the GOE. What is more, and consistent with the above, my own efforts to understand the evolution of reaction centre proteins as a function of time have suggested that the L and M duplication postdate the D1 and D2 duplication; and that the divergence of PshA and PscA postdate the duplication of PsaA and PsaB. Therefore, at the present moment, there is no evidence that clearly demonstrates that anoxygenic photosynthesis ever existed in the absence of oxygenic photosynthesis.

If neither molecular evolution nor the geochemical record have conclusively proven that anoxygenic photosynthesis pre-dates oxygenic photosynthesis, where does the confidence that this is the case come from? I believe this confidence represents an historical and interpretative bias resulting from early speculation on the evolution of photosynthesis, enduring until this day and blurred into the appearance of facts by the inevitable passage of time [14,17–20,124–127].

Nevertheless, there is a direct and easy way to demonstrate that the idea that anoxygenic photosynthesis gave rise to oxygenic photosynthesis is based on unproven assumptions. Oxygenic photosynthesis is characterized by the use of a Type II and a Type I reaction centre linked in series. Regardless of whether these two reaction centres got together via an ancient gene duplication or via horizontal gene transfer, all proposed models for the emergence of oxygenic photosynthesis need to fulfil one major requirement. Namely, that at some point in time a transitional photosynthetic stage using an anoxygenic Type II and a Type I reaction centre was favoured over anoxygenic photosynthesis using a single reaction centre. There is no evidence that such a stage in the evolution of anoxygenic photosynthesis ever existed. There are no described anoxygenic phototrophs with genomes encoding both Type I and anoxygenic Type II reaction centre proteins. If towards the evolution of oxygenic photosynthesis such a transitional stage proved advantageous over 'single-reaction-centre' anoxygenic

photosynthesis, how is it that 'two-reaction-centre' anoxygenic photosynthesis has not evolved several times? In theory, it would only require the transfer of a single gene from a bacterium having a homodimeric Type I reaction centre into one having a Type II. That such a type of anoxygenic photosynthesis is not more common than conventional single-reaction-centre anoxygenic photosynthesis is paradoxical considering that horizontal gene transfer of photosynthesis is supposed to be a fairly feasible process, that anoxygenic photosynthesis is supposed to have emerged in the early Archaean, [128–131], and that most groups of anoxygenic phototrophs have cohabited in microbial mats for billions of years [41,132]. One could argue that there is a functional barrier preventing the acquisition and integration of a second anoxygenic photosystem.

Frankly, observations from the natural world do not match the expectations derived from current models on the evolution of photosynthesis. Consequently, to prove that anoxygenic indeed pre-dates oxygenic photosynthesis one must first demonstrate that 'two-reaction-centre' anoxygenic photosynthesis can provide a competitive advantage or increased fitness over 'single-reaction-centre' anoxygenic photosynthesis. If one can prove that, then one must explain why this two-reaction-centre anoxygenic photosynthesis did not outcompete and supersede the only known form of anoxygenic photosynthesis across all photic ecosystems. This has not been done yet.

There is one way out of this paradox: that photochemical water oxidation originated before or at the divergence of Type I and Type II reaction centres.

Three basic observations from the natural world straightforwardly indicate that the origin of reaction centres was intimately linked to the origin of photochemical water oxidation. These are: (i) that oxygenic photosynthesis is the only process that exclusively uses both reaction centres; (ii) that anoxygenic photosynthesis using both reaction centres does not exist; and (iii) that Photosystem II, the water oxidizing enzyme, is a chimeric photosystem made of a Type II core bound to a Type I antenna with both parts needed for the coordination of the $Mn_4CaO_5$ cluster [133]. Given these three observations, a better starting hypothesis in the study of evolution of photosynthesis is that the origin of photochemical reaction centres was linked to the origin of water oxidation to oxygen, rather than to the origin of a speculative form of anoxygenic photosynthesis.

There is a fourth observation that had until recently remained unnoticed, yet it is straightforward nonetheless: that Photosystem II is the slowest evolving of all photosystems, evolving at a fifth of the rate of anoxygenic Type II reaction centres [34], and about a third of the rate of Type I reaction centres [35]. That means that Photosystem II is the most likely photosystem to have retained traits once found in the most ancestral reaction centre (figures 1 and 2). Strong support for this superior starting hypothesis was revealed in the recent structure of the homodimeric Type I reaction centre from Heliobacteria [45]. It showed a calcium bound at the electron donor site of the core with unmistakable structural parallels to the $Mn_4CaO_5$ cluster of Photosystem II (figure 1) [134]. These structural parallels indicate that the ancestral reaction centre before Type II and Type I had, at the very least, all of the structural elements already in place for the evolution of the $Mn_4CaO_5$ cluster. This is because the most recent common ancestor of Photosystem II and the homodimeric Type I reaction centres is the most recent common ancestor of all reaction centres.

# 5. Final words

The study of the origin and evolution of photosynthesis has fascinated scientists for decades and will continue to capture the imagination of the scientific community and the public for decades to come. It is too soon to claim that we understand how photosynthesis originated, let alone to claim that we understand the photochemistry of the earliest reaction centres to ascertain that the origin of anoxygenic photosynthesis pre-dates the origin of oxygenic photosynthesis. For the field to move forward unhindered, more critical, cautious, yet open thought is required. The temptation to speculate will always be too sweet to resist; nonetheless, we should strive to keep the lines between assumptions, hypotheses, predictions and facts well delimited. Only then we can lay new foundations upon which to build a modern framework for the study of the evolution of photosynthesis.

Data accessibility. This article has no additional data.

Competing interests. I declare I have no competing interests.

Funding. This work was funded by the Leverhulme Trust (grant no. RPG-2017-223).

Acknowledgements. I wish to thank Carmel Cardona for proofreading the manuscript. I also wish to thank Prof. Anthony W. Larkum and Thomas Oliver for interesting discussions on the topic and additional proofreading. I am grateful to Prof. A. W. Rutherford and Dr Andrea Fantuzzi for extensive and stimulating debates. Additional thanks to Prof. Christoph Heubeck and Dr Martin Homann for useful commentary on the preprint version of this article.

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
