## [Reviewer comments · Open Biology]

Review History

RSOB-18-0246.R0 (Original submission)

Review form: Reviewer 1

Recommendation

Accept with minor revision (please list in comments)

Are each of the following suitable for general readers?

- a) **Title**
Yes
- b) **Summary**
Yes
- c) **Introduction**
Yes

Is the length of the paper justified?

Yes

Should the paper be seen by a specialist statistical reviewer?

No

Is it clear how to make all supporting data available?

Yes

Is the supplementary material necessary; and if so is it adequate and clear?

Not Applicable

Do you have any ethical concerns with this paper?

No

Comments to the Author

Let me start by saying that I profoundly enjoyed reading this ms. Having followed research on the evolution of photosynthesis for several decades and having witnessed a plethora of mutually contradictory evolutionary scenarios come and, unfortunately, rather than go, entrench themselves, it is extremely refreshing to take a step back and see these “well-argued and convincing” hypotheses as what they really are: wild speculations! As rightfully pointed out by the author, this is because, in the absence of sufficient data, we tend to make up strings of a priori arguments generating apparently logical houses of cards. I fully agree with the author that truly progressing in this research field necessitates, apart from more empirical information, first and for all a fundamental questioning of the prejudiced logic put forward to stabilize the old houses of cards. Fittingly, the author entitled his piece as a variation to the heading of John Olson’s 2004 article; the spirit of the present paper is very much in line with John’s critical and non-conformist way of thinking.

I therefore consider this article as perfectly suited for publication in Open Biology, apart from a few passages which I will discuss in the following.

1 My main concern is with the top half of page 11, where the author tries to argue for an O₂-replete Archaean. While he is rightfully critical of flimsy arguments with respect to the evolution of photosynthesis, here he does exactly what he accuses his colleagues of. He cites a string of shaky papers suggesting O₂ at the times of the LUCA. The Weiss paper has since its publication become infamous and its methodological flaws have been pointed out in at least 10 more recent articles. The phylogenetic evidences for pre-LUCA oxidases are indeed “not without controversy” (which is quite an understatement) but, what the heck, let’s cite them nevertheless if they go my way ... The oxygen-tolerant enzymes may be quite old (and pre-date cyanos) but to my understanding nobody ever claimed them to be as old as the LUCA. And if the hypothesis in the Ettwig et al paper on O₂-production from disproportionation of NO in anaerobic methanotrophy is correct, that may explain why SODs and consortia have evolved relatively early (not to mention that nitrosylating stress is very similar to ROS-stress and these enzymes may initially have detoxified nitrosyl moieties).

I am aware that the author wants to leave the door open to the presence of a water-splitting RC already in the LUCA. However, this doesn’t require stipulating an O₂-replete environment since (inefficient) early O₂-evolution by an ancestral RC can have gone on for eons before they had a perceptible impact on global oxygenation. So I don’t see why the author builds on extremely controversial articles to argue against a solid mountain of palaeogeochemical evidences arguing for an O₂-free Hadean and Archaean.

2. The arguments in the lower half of page 11 to my mind don’t hold either. The author uses the absence of anoxygenic 2-RC systems as evidence against anoxygenic-first scenarios. Here is my take on that: Unless one believes that life started photosynthetically (I am not sure whether the author envisages such beginnings but I certainly don’t!), photosynthetic RCs had to plug

themselves into pre-existing electron transfer chains. If you look at the positioning of RCs in these chains (be they of RCI or RCII type), it is obvious that they replace terminal oxidants. This makes perfect sense since (prior to ubiquitous O₂ in the environment) oxidants were less abundant than reductants and more readily limiting. Both RCII and RCI do this job and they even do it ok when strongly reducing substrates become scarce and bugs are left to resort to the weakly reducing ones. RCI might be more efficient under these conditions since it can reduce NAD directly without having to tap into the membrane potential to make reverse electron transfer go.

However, if a bug dares to venture out into the open waters, far from hydrothermal sources of reductants, it will get knackered – unless it manages to use water as a reductant, that is, invents a water-oxidising RC. Note that it will still require the anoxygenic photosynthetic oxidant to ensure pmf-generating electron flow. So, if you look at photosynthesis as life's way to become energetically independent of Mother Earth's redox energy by tapping into heavenly sources of entropic disequilibria, then it is obvious that an anoxygenic RC will be the first step (replacing terminal oxidants) but going all the way to the middle of the ocean necessitates an oxygenic RC replacing the reductants AND an anoxygenic RC replacing the oxidants. By contrast, two anoxygenic RCs in the same chain wouldn't make sense since they do the same job (just consider your Gemmatimonadetes where an RCII photosynthesis has simply replaced an RCI-type one).

2. Last paragraph before "final words". While extant life uses maturation systems to insert FeS clusters, these clusters are natural entities. The maturation systems only enhance efficiency. You can unfold most iron-sulphur cluster containing enzymes and just feed them iron chloride and H₂S and when you dialyse out the urea these enzymes will refold and contain native FeS clusters, no maturation proteins needed. This is not the case for the Manganese cluster and the resemblance of its geometry to mineral-type manganese containing entities is quite limited. It is a product of PSII photochemistry and therefore less "natural" than for example iron-sulphur centres.

Typos:

Page 8, line 1: zero

Page 10, line 6: signs

Figure 4: how do you make 4 NADHs out of 4 electrons?

Review form: Reviewer 2

Recommendation

Accept with minor revision (please list in comments)

Are each of the following suitable for general readers?

- a) **Title**
Yes
- b) **Summary**
Yes
- c) **Introduction**
No

Is the length of the paper justified?

Yes

Should the paper be seen by a specialist statistical reviewer?

No

Is it clear how to make all supporting data available?

Not Applicable

Is the supplementary material necessary; and if so is it adequate and clear?

Not Applicable

Do you have any ethical concerns with this paper?

No

Comments to the Author

The ideas summarized here are interesting and thought provoking and the data reviewed here from Cardona and colleagues has already been peer-reviewed for the original publications.

The purpose of the present document therefore appears to be to bring the new ideas from the author to a wider audience who might be more familiar with earlier entrenched ideas regarding the origin of oxygenic photosynthesis and the distribution of photosynthesis.

The target audience for this article is important since the author has adopted a rather informal style e.g.,

“..... and [a] few years before I took the study of the evolution of photosynthesis as my full-time job I came across an analogy that left me flabbergasted...”

“...our best scenarios for the origin and diversification of photosynthesis are like trying to predict the size of a mountain from the shape of a [leaf] of grass...”

“What about other independent lines of evidence? Any clues?”

“How about early evolving oxygen-tolerant hydrogenases...substantially predate the mrca of Cyanobacteria?”

The entire paragraph beginning “If neither molecular evolution nor the geochemical record...”

The italics for emphasis in the statement: “There is absolutely no evidence that such a stage in the evolution of anoxygenic photosynthesis ever existed”

The choice of the word “bizarre”

The above and a number of other examples risk, in my view, alienating others in the field who have made (and are making) major contributions based on the evidence available to them and who have contributed their own thought provoking ideas in the past (and quite recently in several cases).

The author most likely intends no offence and other slip ups in language suggest the nuances inherent in some of the criticisms of previous or alternative views may not be intentional but, nevertheless, they do run the risk of coming across as arrogant or condescending. I suggest an edit of the document, using a native speaker who is knowledgeable in photosynthesis, to remove the more emotive language.

I enjoyed the scientific content and hold the author’s contributions in high regard.

Decision letter (RSOB-18-0246.R0)

06-Feb-2019

Dear Dr Cardona

We are pleased to inform you that your manuscript RSOB-18-0246 entitled "Thinking Twice about the Evolution of Photosynthesis" has been accepted by the Editor for publication in Open Biology. The reviewer(s) have recommended publication, but also suggest some minor revisions to your manuscript. Therefore, we invite you to respond to the reviewer(s)' comments and revise your manuscript.

Please submit the revised version of your manuscript within 14 days. If you do not think you will be able to meet this date please let us know immediately and we can extend this deadline for you.

- 1) A text file of the manuscript (doc, txt, rtf or tex), including the references, tables (including captions) and figure captions. Please remove any tracked changes from the text before submission. PDF files are not an accepted format for the "Main Document".
- 2) A separate electronic file of each figure (tiff, EPS or print-quality PDF preferred). The format should be produced directly from original creation package, or original software format. Please note that PowerPoint files are not accepted.
- 3) Electronic supplementary material: this should be contained in a separate file from the main text and meet our ESM criteria (see <http://royalsocietypublishing.org/instructions-authors#question5>). All supplementary materials accompanying an accepted article will be treated as in their final form. They will be published alongside the paper on the journal website and posted on the online figshare repository. Files on figshare will be made available approximately one week before the accompanying article so that the supplementary material can be attributed a unique DOI.

Online supplementary material will also carry the title and description provided during submission, so please ensure these are accurate and informative. Note that the Royal Society will not edit or typeset supplementary material and it will be hosted as provided. Please ensure that

the supplementary material includes the paper details (authors, title, journal name, article DOI). Your article DOI will be 10.1098/rsob.2016[*last 4 digits of e.g. 10.1098/rsob.20160049*].

4) A media summary: a short non-technical summary (up to 100 words) of the key findings/importance of your manuscript. Please try to write in simple English, avoid jargon, explain the importance of the topic, outline the main implications and describe why this topic is newsworthy.

Images

Data-Sharing

It is a condition of publication that data supporting your paper are made available. Data should be made available either in the electronic supplementary material or through an appropriate repository. Details of how to access data should be included in your paper. Please see <http://royalsocietypublishing.org/site/authors/policy.xhtml#question6> for more details.

Sincerely,

The Open Biology Team
<mailto:openbiology@royalsociety.org>

Reviewer(s)' Comments to Author:

Referee: 1

Comments to the Author(s)

Let me start by saying that I profoundly enjoyed reading this ms. Having followed research on the evolution of photosynthesis for several decades and having witnessed a plethora of mutually contradictory evolutionary scenarios come and, unfortunately, rather than go, entrench themselves, it is extremely refreshing to take a step back and see these “well-argued and convincing” hypotheses as what they really are: wild speculations! As rightfully pointed out by the author, this is because, in the absence of sufficient data, we tend to make up strings of a priori arguments generating apparently logical houses of cards. I fully agree with the author that truly progressing in this research field necessitates, apart from more empirical information, first and for all a fundamental questioning of the prejudiced logic put forward to stabilize the old houses of cards. Fittingly, the author entitled his piece as a variation to the heading of John Olson’s 2004 article; the spirit of the present paper is very much in line with John’s critical and non-conformist way of thinking.

I therefore consider this article as perfectly suited for publication in Open Biology, apart from a few passages which I will discuss in the following.

1 My main concern is with the top half of page 11, where the author tries to argue for an O₂-replete Archaean. While he is rightfully critical of flimsy arguments with respect to the evolution of photosynthesis, here he does exactly what he accuses his colleagues of. He cites a string of shaky papers suggesting O₂ at the times of the LUCA. The Weiss paper has since its publication become infamous and its methodological flaws have been pointed out in at least 10 more recent articles. The phylogenetic evidences for pre-LUCA oxidases are indeed “not without

controversy” (which is quite an understatement) but, what the heck, let’s cite them nevertheless if they go my way ... The oxygen-tolerant enzymes may be quite old (and pre-date cyanos) but to my understanding nobody ever claimed them to be as old as the LUCA. And if the hypothesis in the Ettwig et al paper on O₂-production from disproportionation of NO in anaerobic methanotrophy is correct, that may explain why SODs and consortia have evolved relatively early (not to mention that nitrosylating stress is very similar to ROS-stress and these enzymes may initially have detoxified nitrosyl moieties).

I am aware that the author wants to leave the door open to the presence of a water-splitting RC already in the LUCA. However, this doesn’t require stipulating an O₂-replete environment since (inefficient) early O₂-evolution by an ancestral RC can have gone on for eons before they had a perceptible impact on global oxygenation. So I don’t see why the author builds on extremely controversial articles to argue against a solid mountain of palaeochemical evidences arguing for an O₂-free Hadean and Archaean.

2. The arguments in the lower half of page 11 to my mind don’t hold either. The author uses the absence of anoxygenic 2-RC systems as evidence against anoxygenic-first scenarios. Here is my take on that: Unless one believes that life started photosynthetically (I am not sure whether the author envisages such beginnings but I certainly don’t!), photosynthetic RCs had to plug themselves into pre-existing electron transfer chains. If you look at the positioning of RCs in these chains (be they of RCI or RCII type), it is obvious that they replace terminal oxidants. This makes perfect sense since (prior to ubiquitous O₂ in the environment) oxidants were less abundant than reductants and more readily limiting. Both RCII and RCI do this job and they even do it ok when strongly reducing substrates become scarce and bugs are left to resort to the weakly reducing ones. RCI might be more efficient under these conditions since it can reduce NAD directly without having to tap into the membrane potential to make reverse electron transfer go.

However, if a bug dares to venture out into the open waters, far from hydrothermal sources of reductants, it will get knackered – unless it manages to use water as a reductant, that is, invents a water-oxidising RC. Note that it will still require the anoxygenic photosynthetic oxidant to ensure pmf-generating electron flow. So, if you look at photosynthesis as life’s way to become energetically independent of Mother Earth’s redox energy by tapping into heavenly sources of entropic disequilibria, then it is obvious that an anoxygenic RC will be the first step (replacing terminal oxidants) but going all the way to the middle of the ocean necessitates an oxygenic RC replacing the reductants AND an anoxygenic RC replacing the oxidants. By contrast, two anoxygenic RCs in the same chain wouldn’t make sense since they do the same job (just consider your Gemmatimonadetes where an RCII photosynthesis has simply replaced an RCI-type one).

2. Last paragraph before “final words”. While extant life uses maturation systems to insert FeS clusters, these clusters are natural entities. The maturation systems only enhance efficiency. You can unfold most iron-sulphur cluster containing enzymes and just feed them iron chloride and H₂S and when you dialyse out the urea these enzymes will refold and contain native FeS clusters, no maturation proteins needed. This is not the case for the Manganese cluster and the resemblance of its geometry to mineral-type manganese containing entities is quite limited. It is a product of PSII photochemistry and therefore less “natural” than for example iron-sulphur centres.

Typos:

Page 8, line 1: zero

Page 10, line 6: signs

Figure 4: how do you make 4 NADHs out of 4 electrons?

Referee: 2

Comments to the Author(s)

The ideas summarized here are interesting and thought provoking and the data reviewed here from Cardona and colleagues has already been peer-reviewed for the original publications.

The purpose of the present document therefore appears to be to bring the new ideas from the author to a wider audience who might be more familiar with earlier entrenched ideas regarding the origin of oxygenic photosynthesis and the distribution of photosynthesis.

The target audience for this article is important since the author has adopted a rather informal style e.g.,

“..... and [a] few years before I took the study of the evolution of photosynthesis as my full-time job I came across an analogy that left me flabbergasted...”

“...our best scenarios for the origin and diversification of photosynthesis are like trying to predict the size of a mountain from the shape of a [leaf] of grass...”

“What about other independent lines of evidence? Any clues?”

“How about early evolving oxygen-tolerant hydrogenases...substantially predate the mrca of Cyanobacteria?”

The entire paragraph beginning “If neither molecular evolution nor the geochemical record...”

The italics for emphasis in the statement: “There is absolutely no evidence that such a stage in the evolution of anoxygenic photosynthesis ever existed”

The choice of the word “bizarre”

The above and a number of other examples risk, in my view, alienating others in the field who have made (and are making) major contributions based on the evidence available to them and who have contributed their own thought provoking ideas in the past (and quite recently in several cases).

The author most likely intends no offence and other slip ups in language suggest the nuances inherent in some of the criticisms of previous or alternative views may not be intentional but, nevertheless, they do run the risk of coming across as arrogant or condescending. I suggest an edit of the document, using a native speaker who is knowledgeable in photosynthesis, to remove the more emotive language.

I enjoyed the scientific content and hold the author’s contributions in high regard.

Author's Response to Decision Letter for (RSOB-18-0246.R0)

See Appendix A.

Decision letter (RSOB-18-0246.R1)

25-Feb-2019

Dear Dr Cardona

We are pleased to inform you that your manuscript entitled "Thinking Twice about the Evolution of Photosynthesis" has been accepted by the Editor for publication in Open Biology.

Sincerely,

The Open Biology Team
mailto: openbiology@royalsociety.org

Appendix A

Referee: 1

Comments to the Author(s)

Let me start by saying that I profoundly enjoyed reading this ms. Having followed research on the evolution of photosynthesis for several decades and having witnessed a plethora of mutually contradictory evolutionary scenarios come and, unfortunately, rather than go, entrench themselves, it is extremely refreshing to take a step back and see these “well-argued and convincing” hypotheses as what they really are: wild speculations! As rightfully pointed out by the author, this is because, in the absence of sufficient data, we tend to make up strings of a priori arguments generating apparently logical houses of cards. I fully agree with the author that truly progressing in this research field necessitates, apart from more empirical information, first and for all a fundamental questioning of the prejudiced logic put forward to stabilize the old houses of cards. Fittingly, the author entitled his piece as a variation to the heading of John Olson’s 2004 article; the spirit of the present paper is very much in line with John’s critical and non-conformist way of thinking.

I therefore consider this article as perfectly suited for publication in Open Biology, apart from a few passages which I will discuss in the following.

1. My main concern is with the top half of page 11, where the author tries to argue for an O₂-replete Archaean. While he is rightfully critical of flimsy arguments with respect to the evolution of photosynthesis, here he does exactly what he accuses his colleagues of. He cites a string of shaky papers suggesting O₂ at the times of the LUCA. The Weiss paper has since its publication become infamous and its methodological flaws have been pointed out in at least 10 more recent articles. The phylogenetic evidences for pre-LUCA oxidases are indeed “not without controversy” (which is quite an understatement) but, what the heck, let’s cite them nevertheless if they go my way ... The oxygen-tolerant enzymes may be quite old (and pre-date cyanos) but to my understanding nobody ever claimed them to be as old as the LUCA. And if the hypothesis in the Ettwig et al paper on O₂-production from disproportionation of NO in anaerobic methanotrophy is correct, that may explain why SODs and consortia have evolved relatively early (not to mention that nitrosylating stress is very similar to ROS-stress and these enzymes may initially have detoxified nitrosyl moieties).

I am aware that the author wants to leave the door open to the presence of a water-splitting RC already in the LUCA. However, this doesn’t require stipulating an O₂-replete environment since (inefficient) early O₂-evolution by an ancestral RC can have gone on for eons before they had a perceptible impact on global oxygenation. So I don’t see why the author builds on extremely controversial articles to argue

against a solid mountain of palaeo-geochemical evidences arguing for an O₂-free Hadean and Archean.

I thank the reviewer for pointing this out and discussing the topic. I think I did not make myself clear. I am not trying to argue for an O₂ replete Archean: not at all. I am trying to argue that neither molecular evolution data nor geochemical data have been able to time the origin of oxygenic photosynthesis conclusively and to rule out that photosynthetically derived O₂ was not driving the evolution of early bioenergetics and early life. *Even if O₂ did not accumulate in the atmosphere until the GOE.*

I was also trying to say that both molecular evolution data and geochemical data have been interpreted to suggest availability of O₂ to life, and even possibly oxygenic photosynthesis, as early as the earliest evidence for life. That does not necessarily imply an O₂ replete Archean.

In other words, my intention was to contrast the ambiguity of the data with the confidence placed in the premise that anoxygenic photosynthesis predates oxygenic photosynthesis.

I strongly agree with geochemical evidence for a very low-oxygen world prior to the GOE, I want to make this clear: I do not dispute this. It is my understanding however, that even though O₂ was not accumulating globally there is substantial evidence for O₂ oases/whiffs or fluctuations of O₂ long before the GOE.

In the words of Wang, Planavsky, Hofmann *et al.* (2018): “*Further, not all pre-2.4 Ga rocks have MIF-S signals, suggesting that oxygen levels likely fluctuated dramatically during the Archean Era (Ohmoto et al., 2006, Ono et al., 2006). With a small O₂ reservoir, atmospheric oxygen levels were likely highly dynamic, with oxygen concentrations varying significantly on short time scales.*”

I want to restate that I am not using the controversial articles on the molecular evolution of O₂ reductase or the LUCA reconstructions to argue against geochemical evidence. I only use them to point out that the interpretations of the molecular evolution of O₂ reductases and ROS-handling enzymes have been inconclusive on the matter of when O₂ becomes available to life for the first time.

The other point that I want to make is that I am not using potential evidence of early O₂ as an argument supporting the early origin of oxygenic photosynthesis. It is the opposite:

The evolution of reaction centres suggests an early origin of oxygenic photosynthesis irrespective of whether there were abiotic or other biological sources of O₂; and irrespective of the geological process that controlled the atmospheric composition during the Archean. Therefore, given that the evolution of reaction centres indicates an early origin of oxygenic photosynthesis, then evidence for early oxygen may be better explained in the context of photosynthesis.

I have made the following changes to make my points clearer. The paragraph on geochemical and palobiological evidence starts like this: *“From a paleobiological and geochemical perspective, the timing of the origin of oxygenic photosynthesis is far from settled.”* I have changed the start of the next paragraph on evidence from molecular evolution to give more context on the point that I want to make: *“Molecular evolution is equally inconclusive on the timing of the origin of biological oxygen production.”* I have also added a short paragraph considering molecular clock studies on prokaryotes, all of which suggest late origins for the known groups of anoxygenic phototrophs. I have also made minor editions on the entire “Let there be light” section to make my points clearer and following reviewer-2’s recommendations.

2. The arguments in the lower half of page 11 to my mind don’t hold either. The author uses the absence of anoxygenic 2-RC systems as evidence against anoxygenic-first scenarios. Here is my take on that: Unless one believes that life started photosynthetically (I am not sure whether the author envisages such beginnings but I certainly don’t!), photosynthetic RCs had to plug themselves into pre-existing electron transfer chains. If you look at the positioning of RCs in these chains (be they of RCI or RCII type), it is obvious that they replace terminal oxidants. This makes perfect sense since (prior to ubiquitous O₂ in the environment) oxidants were less abundant than reductants and more readily limiting. Both RCII and RCI do this job and they even do it ok when strongly reducing substrates become scarce and bugs are left to resort to the weakly reducing ones. RCI might be more efficient under these conditions since it can reduce NAD directly without having to tap into the membrane potential to make reverse electron transfer go. However, if a bug dares to venture out into the open waters, far from hydrothermal sources of reductants, it will get knackered – unless it manages to use water as a reductant, that is, invents a water-oxidising RC. Note that it will still require the anoxygenic photosynthetic oxidant to ensure pmf-generating electron flow. So, if you look at photosynthesis as life’s way to become energetically independent of Mother Earth’s redox energy by tapping into heavenly sources of entropic disequilibria, then it is obvious that an anoxygenic RC will be the first step (replacing terminal oxidants) but going all the way to the middle of the ocean necessitates an oxygenic RC replacing the reductants AND an anoxygenic RC replacing the oxidants. By contrast, two anoxygenic RCs in the same chain wouldn’t make sense since they do the same job (just consider your Gemmatimonadetes where an RCII photosynthesis has simply replaced an RCI-type one).

The reviewer says that my arguments about the lack of anoxygenic 2-RC systems don’t hold, but I think the reviewer actually agrees with me at the end. I would like to comment on a few of the points made by the reviewer.

1. The reviewer wrote: *“The author uses the absence of anoxygenic 2-RC systems as evidence against anoxygenic-first scenarios.”*

This is not quite the case. I use the absence of anoxygenic 2-RC systems to demonstrate that the idea that anoxygenic photosynthesis predates oxygenic photosynthesis is based on unsupported assumptions. Then, I argued that if the origin

of oxygenic photosynthesis is coincidental with the divergence of Type I and Type II reaction centres, as the data actually suggests, then one does not need to invoke this unsupported assumption to make sense of the natural history of photosynthesis.

2. *“(I am not sure whether the author envisages such beginnings but I certainly don’t!)”*

I must say that I have started to find the idea more and more appealing. Unfortunately, I am not an expert on Origin of Life scenarios. Yet, Sam Granick in his 1957 paper imagined an oxygenic photosynthetic origin of life. Quoting: *“It seems more reasonable to consider that the functions of oxidation and photosynthesis were so fundamental that they were part of the first beginnings of protoplasm that arose from inorganic origins.”* Then he goes on to say: *“I propose, as speculation, that the earliest unit around which any living entity arose was an energy-conversion unit. This unit of mineral origin would contain an organization of atoms that would serve as a photocatalyst, at first perhaps in the decomposition of water by UV radiation.”*

However, it is not about whether one can envision such beginnings. It is about the data itself.

My approach to understand the evolution of reaction centres has always been to separate the speculation from those conclusions that are supported by data.

When I arrived to the field, I had no reason to doubt the starting assumption that anoxygenic photosynthesis predates oxygenic photosynthesis. However, when I first examined the evolution of reaction centres the first thing that I noticed is that the relationship between the reaction centre proteins indicated, and I dare to say rather conclusively, that the divergence of Type I and Type II RC proteins predates the diversification of the major groups of bacteria (Cardona 2015). I also noted that the phylogeny of the RC proteins quite puzzlingly showed that the cyanobacterial sequences did not emerge from within the anoxygenic sequences (as it is expected under traditional views), but made completely independent clades branching out at the root.

After that, I tried to understand why Photosystem II is structurally and functionally the way it is. The data suggest that in order to account for the structural characteristics of Photosystem II and its oxygen-evolving complex, it is necessary that water oxidation started at the evolutionary event that led to the two known types of photochemical reaction centres (Cardona 2017, Cardona and Rutherford 2019, Cardona *et al.* 2019). This is not my speculation, this is strongly constrained by the structural characteristics of Photosystem II itself. The new structure of the Heliobacteria RC only provided further support. These considerations consequently place the origin of water oxidation at an early stage within the evolution of Bacteria, at the latest. These conclusions are also consistent with the independent observation that the span of time between the duplication of D1 and D2 and the most recent common ancestor of cyanobacteria can be easily more than a billion years.

The reason I feel somewhat confident about this is because it has tremendous explanatory power. Not only it explains why Photosystem II is the way it is, starting from the curious ligand sphere of the Mn_4CaO_5 cluster and ending with its entire structural architecture (Cardona 2017, Cardona and Rutherford 2019), but also it equally explains why the phylogenies of RC proteins have the topology they do, and it also explains the fact that there is no anoxygenic photosynthesis using the two types of reaction centres.

I have now become very interested in the evolutionary parallels between (F/V-type) ATP synthase and Photosystem II. I have found that across any point in time it looks like Photosystem II is evolving more slowly than ATP synthase. This relationship is even maintained through primary endosymbiosis of cyanobacteria at the origin of photosynthetic eukaryotes. What is more, CP43 and CP47 are evolving slightly slower than the alpha and beta subunits of ATP synthase, with D1 and D2 evolving considerably slower; with the distance between alpha and beta being almost identical to that between D1 and D2, and between CP43 and CP47. This is unpublished data, but it can be easily confirmed by just doing a few sequence comparisons and as I illustrated in Figure 4. I find no evidence whatsoever to suggest that the gene duplication event leading to alpha and beta subunit of ATP synthase, a pre-LUCA event, occurred before the duplications leading to D1 and D2 or CP43 and CP47, **as it is expected under the traditional view**. The data taken at face-value appear to suggest the opposite, that the D1/D2 and CP43/47 duplications are coincidental or may slightly predate the duplication leading to alpha and beta. I think this deserves further investigation and I am currently trying to get additional data on this.

But as I mentioned above, I had already shown that the divergence of Type I and Type II reaction centre proteins likely predates the diversification of the major groups of bacteria. I recently showed that the divergence of Type I and II, the divergence of the L/M and D1/D2 lineages, and the duplication of D1 and D2, likely occurred soon after the origin of reaction centre proteins (Cardona *et al.* 2019). Therefore, I do not find puzzling or strange the fact that the evolutionary patterns of Photosystem II are in many ways indistinguishable from those of ATP synthase. In fact, that is exactly what one should expect.

Because of the above, I have become open-minded about the possibility of water oxidation chemistry being important for the origin and evolution of bioenergetics and the possibility of it emerging at any point in time between the Origin of Life and the early diversification of Bacteria. I want to remind the reviewer that the spans of time between the Origin of Life and the LUCA, and between the LUCA and the early diversification of Bacteria are unknown.

The main issue, of course, is the fact that photosynthesis is restricted to few groups of bacteria, and oxygenic photosynthesis to cyanobacteria alone. At a superficial look it would seem as if photosynthesis was definitely a late innovation, and this has been the only interpretative framework that we have had until now. However, the nature of reaction centres and the chlorophyll synthesis pathway tell a completely different story (as I have tried to make evident in this critical review): a story that is obscured

by the old interpretative framework. Once this framework is dismantled, a simpler and consistent story with more explanatory power emerges. I hope at some point in the not too distant future to have time and funding to write a dedicated and detailed book on the evolution of photosynthesis.

3. The reviewer wrote: *“If you look at the positioning of RCs in these chains (be they of RCI or RCII type), it is obvious that they replace terminal oxidants.”*

This is only obvious within the traditional interpretative framework of the evolution of photosynthesis, which as I have tried to argue, is very problematic and biased towards the idea that anoxygenic photosynthesis is the more plausible ancestral state. While it appears reasonable, there is no support for it.

4. *“However, if a bug dares to venture out into the open waters, far from hydrothermal sources of reductants, it will get knackered – unless it manages to use water as a reductant, that is, invents a water-oxidising RC. “*

If that is correct, it would imply an early origin of oxygenic photosynthesis and soon after the origin of life. As far as I understand, life was thriving in the Archean and was not exclusively bound to hydrothermal environments, see for example (Homann *et al.* 2015). So according to what the reviewer wrote, in order for life to leave its cradle, it would need oxygenic photosynthesis. This would place oxygenic photosynthesis at a very early stage in the evolutionary history of life. I think the reviewer agrees with me.

5. *“By contrast, two anoxygenic RCs in the same chain wouldn’t make sense since they do the same job”*

Exactly! So, there’s a natural barrier that prevent this stage from emerging and therefore cannot be considered a valid transitional stage towards the evolution of oxygenic photosynthesis.

For a two-RC anoxygenic system to have emerged it needed to have provided a selective advantage over single-RC anoxygenic system. If that would have been the case, one would expect multiple events of two-RC anoxygenic systems emerging. In fact, over billions of years, one would expect that it superseded single-RC anoxygenic systems. But this clearly has not happened, which is indeed paradoxical.

The *“two anoxygenic RCs in the same chain”* is a paradoxical evolutionary transition towards the origin of oxygenic photosynthesis, and necessitates the incorporation of unproven and unsupported assumptions, which is the exact point I want to make.

I have made the following changes to make my points clearer. I added a paragraph in page 7 of the revised manuscript highlighting many independent molecular clock studies of prokaryotes. These place each of the *most recent common ancestors* of

each group of phototroph at a late stage (e.g. about the GOE or after the GOE). It further highlights that there is no evidence that anoxygenic photosynthesis has ever existed in the absence of oxygenic photosynthesis. In addition, in the following paragraph (start of page 8), I added a few sentences about what needs to be done to prove conclusively that anoxygenic photosynthesis predate oxygenic photosynthesis, which should help make things clearer regarding the current gaps in knowledge.

3. Last paragraph before “final words”. While extant life uses maturation systems to insert FeS clusters, these clusters are natural entities. The maturation systems only enhance efficiency. You can unfold most iron-sulphur cluster containing enzymes and just feed them iron chloride and H₂S and when you dialyse out the urea these enzymes will refold and contain native FeS clusters, no maturation proteins needed. This is not the case for the Manganese cluster and the resemblance of its geometry to mineral-type manganese containing entities is quite limited. It is a product of PSII photochemistry and therefore less “natural” than for example iron-sulphur centres.

I would like to thank the reviewer for pointing this out. It is indeed a weak argument in the absence of a more detailed discussion of the potential implications of the photoactivation process of the Mn₄CaO₅ cluster on the evolution of water oxidation. I have decided to delete this paragraph.

Typos:

Page 8, line 1: zero

Page 10, line 6: signs

Figure 4: how do you make 4 NADHs out of 4 electrons?

These have been corrected. Thanks.

Referee: 2

Comments to the Author(s)

The ideas summarized here are interesting and thought provoking and the data reviewed here from Cardona and colleagues has already been peer-reviewed for the original publications.

The purpose of the present document therefore appears to be to bring the new ideas from the author to a wider audience who might be more familiar with earlier entrenched ideas regarding the origin of oxygenic photosynthesis and the distribution of photosynthesis.

The target audience for this article is important since the author has adopted a rather informal style e.g.,

“..... and [a] few years before I took the study of the evolution of photosynthesis as my full-time job I came across an analogy that left me flabbergasted...”

"...our best scenarios for the origin and diversification of photosynthesis are like trying to predict the size of a mountain from the shape of a [leaf] of grass..."

"What about other independent lines of evidence? Any clues?"

"How about early evolving oxygen-tolerant hydrogenases...substantially predate the mrca of Cyanobacteria?"

The entire paragraph beginning "If neither molecular evolution nor the geochemical record..."

The italics for emphasis in the statement: "There is absolutely no evidence that such a stage in the evolution of anoxygenic photosynthesis ever existed"

The choice of the word "bizarre"

The above and a number of other examples risk, in my view, alienating others in the field who have made (and are making) major contributions based on the evidence available to them and who have contributed their own thought provoking ideas in the past (and quite recently in several cases).

The author most likely intends no offence and other slip ups in language suggest the nuances inherent in some of the criticisms of previous or alternative views may not be intentional but, nevertheless, they do run the risk of coming across as arrogant or condescending. I suggest an edit of the document, using a native speaker who is knowledgeable in photosynthesis, to remove the more emotive language.

I enjoyed the scientific content and hold the author's contributions in high regard.

I want to thank the reviewer for highlighting these issues of style. I do not mean any offence and I do not intend to alienate others in the field. I did want to use a more informal style to reach a broader audience and to make the article a more entertaining read, I tried my best to not compromise in accuracy. I have edited all instances highlighted by the reviewer above. I also asked my collaborator and friend, Prof. Anthony Larkum to proof-read the manuscript as suggested by the reviewer. I hope these changes make the manuscript more suitable for publication.

Cardona, T. (2015). "A fresh look at the evolution and diversification of photochemical reaction centers." *Photosynth Res* **126**(1): 111-134. DOI: 10.1007/s11120-014-0065-x.

Cardona, T. (2017). "Photosystem II is a chimera of reaction centers." *J Mol Evol* **84**(2-3): 149-151. DOI: 10.1007/s00239-017-9784-x.

- Cardona, T. and A. W. Rutherford (2019). "Evolution of photochemical reaction centres: more twists?" BioRxiv. DOI: <https://doi.org/10.1101/502450>.
- Cardona, T., P. Sanchez-Baracaldo, A. W. Rutherford and A. W. D. Larkum (2019). "Early Archean origin of Photosystem II." Geobiology **17**(2): 127-150. DOI: 10.1111/gbi.12322.
- David, L. A. and E. J. Alm (2011). "Rapid evolutionary innovation during an Archaean genetic expansion." Nature **469**(7328): 93-96. DOI: 10.1038/Nature09649.
- Homann, M., C. Heubeck, A. Airo and M. M. Tice (2015). "Morphological adaptations of 3.22 Ga-old tufted microbial mats to Archean coastal habitats (Moodies Group, Barberton Greenstone Belt, South Africa)." Precambrian Res **266**: 47-64. DOI: 10.1016/j.precamres.2015.04.018.
- Wang, X. L., N. J. Planavsky, A. Hofmann, E. E. Saupe, B. P. De Corte, P. Philippot, S. V. LaLonde, N. E. Jemison, H. J. Zou, F. O. Ossa, K. Rybacki, N. Alfimova, M. J. Larson, H. Tsikos, P. W. Fralick, T. M. Johnson, A. C. Knudsen, C. T. Reinhard and K. O. Konhauser (2018). "A Mesoarchean shift in uranium isotope systematics." Geochim Cosmochim Acta **238**: 438-452. DOI: 10.1016/j.gca.2018.07.024.